# How Does Experiential Value toward Robot Barista Service Affect Emotions, Storytelling, and Behavioral Intention in the Context of COVID-19?

**Se-Ran Yoo** [1], **Seon-Hee Kim** [2] **and Hyeon-Mo Jeon** [3,*]

1 Graduate School, Kyung Hee University, Seoul 02447, Korea; sel97@khu.ac.kr
2 Department of Tourism Management, Gachon University, Seongnam 13120, Korea; anes12054@naver.com
3 Department of Hotel, Tourism, and Foodservice Management, Dongguk University-Gyeongju, Gyeongju 38066, Korea
* Correspondence: jhm010@dongguk.ac.kr; Tel.: +82-10-6275-4010

**Abstract:** This study aims to examine the antecedents of storytelling and intention to a behavioral robot barista coffee shop (RBCS) by exploring experiential values and emotions. For the analysis, a set of hypotheses was developed and tested based on data collected from 300 customers who had visited a RBCS in South Korea. In the verification, the atmosphere showed the greatest influence on positive emotion, followed by consumer return on investment (CROI). These results mean that atmosphere, CROI, and escapism are important to induce positive emotion and behavioral intention for robot barista coffee shop. This is the first study to examine consumers' experiential value regarding non-face-to-face robot service in the food service industry. This design is different from previous experiential value studies on human services in the hospitality industry. By integrating artificial intelligence and digital innovation into food service, this study broadens the scope of research in consumer behavior, making a significant theoretical contribution to the literature. Furthermore, this study proposes practical implications for sustainable coffee shop management in a COVID-19 environment.

**Keywords:** robot barista; experiential value; emotions; storytelling; behavioral intention; coffee shop

## 1. Introduction

The recent health crisis caused by the COVID-19 pandemic has quickly turned into a financial downturn owing to the restrictions imposed by most countries to control the spread of the infection [1]. Like many other industries in the tourism sector, the restaurant industry has been hit extremely hard by the current pandemic [2]. More than eight-million restaurant employees were laid off or furloughed, and the industry lost $280 billion in sales during the first 13 months of the pandemic [3]. Ordering food at home rather than eating out has become the norm [4]. Consumers tend to avoid crowded places and visit restaurants in small groups, following social distancing guidelines for COVID-19 prevention and control [5]. Consequently, restaurants had to close or operate under severe restrictions; only takeout and delivery services were allowed [1].

As COVID-19 is a stumbling block to sales growth in the coffee shop business in Korea, coffee shop operators are trying to minimize losses by focusing on quarantine [6]. Despite this, dozens of employees and customers at the Starbucks' Paju store in South Korea have been infected [7], which shows the importance of contactless service in the food service industry [4]. The restaurant business plays an essential role in the national economy, even though multiple small- and medium-sized restaurants fail in the first four years of business. Therefore, a strategy is required to keep these businesses financially viable for more than five years [1]. Therefore, it is necessary to discuss the sustainable management of the food service industry in the COVID-19 era.

After the World Health Organization (WHO) formally declared COVID-19 a pandemic in March 2020, the shift to digital shopping via online platforms, which complies with

social distancing measures, became more prominent than ever [8]. Even before the onset of the pandemic, contactless transactions were rapidly accelerated by the integration of advanced AI-based technology, innovative services, and food technology [9]. In particular, smart robots are increasingly replacing workers by providing contactless services during the pandemic [10]. One example is that an increasing number of coffee shops now use robot baristas [9]. In Japan, coffee shops have already initiated the use of humanoid robots named "Pepper PARLOR." In addition to brewing coffee, these robots take orders, entertain customers, and clean. In China, the company OrionStar created a robot barista who can assume the role of a human barista by pouring water from a kettle over ground coffee beans and extracting drip coffee [9]. Similarly, a South Korean robotic coffee shop called "Lounge X" is now using a robot barista named "BARIS CAN," which is able to take orders and serve prepared coffee, thereby pioneering AI-oriented sensitivity and culture for unmanned cafes, as well as providing insight into the future of AI-enabled cafes [11].

Robots are emerging from industrial robots to become a representative service means in the (no contact) untact era [12]. It is expected that the robot service in the hospitality industry will be further expanded as the speed of introduction of automation technology by hospitality companies is accelerated due to the preference for untact services and the convergence of innovative technologies after COVID-19 [13]. Despite this situation, empirical research on robot service experience in the hospitality industry has not been sufficiently conducted. In particular, in the hospitality industry, a study that applied the experience theory to measure the emotional response of customers and the behavior of consumers on the robot barista service was not conducted. Looking at customer emotions research on AI-based technologies and robotic services in the hospitality industry, Gursoy et al. [14] developed and empirically tested a theoretical model of artificial intelligence (AI) device use acceptance (AIDUA), which aims to account for customers' willingness to accept the use of such devices in service encounters. They confirmed that performance expectancy and effort expectancy are important antecedents of customer emotions. Guo et al. [15] investigated the effect of a humanoid robot's emotional behaviors on users' emotional responses using subjective reporting, pupillometry, and electroencephalography. As described above, research on customer emotions regarding AI-based technology and robot services is limited in the hospitality industry.

Experiential value refers to customers' perceived and relativistic preferences for product attributes or service performances that are provided at contact points [16]. Customer experiential value is measurable in terms of both utilitarian (e.g., time and convenience) and hedonic (e.g., feelings of pleasure, escapism, and joy) aspects of consumption, serving as an important factor that influences customer emotions [17]. Experiential value has been steadily studied in the restaurant sector. Particularly, it has been explored in terms of CROI, service excellence, atmosphere, and escapism, among other dimensions, to fit the restaurant sector characteristics that value service encounter with customers [18–21]. Since the robot barista in the coffee shop assumes the role of a human barista in the service encounter context with the customer, experience value theory applied to the human service in the restaurant is judged to be suitable for the robot service context. We therefore apply the four dimensions of experience value in the current study.

Emotions are an emotional reaction or feeling that occurs through the use or consumption of a product or service [22]. Emotions generated via experience recognition have two-sided characteristics [23]. In this context, individual emotional states can be divided into negative states (e.g., annoyed and stressed) and positive states (e.g., happy and comfortable) [24]. For example, when customers feel positive about the family restaurant, they are more likely to become emotionally attached to it, which in turn increases their loyalty to the brand [18,25]. Many researchers have considered emotions as an important driver of service quality evaluation, brand loyalty development, repeat buying behavior, and service provider selection [18,26].

Consumer responses to service experiences shape their affective state, which is then memorized and shared with others in the form of storytelling [27]. Consumers recall their

experience through storytelling and experience pleasurable feelings by sharing it with others through word of mouth (WOM) communication [28,29]. Storytelling can relay positive emotion connected to customer service [30], and eventually affect behavioral intention [27]. Hence, storytelling techniques are widely used in the hospitality industry [31,32].

As such, this study seeks to determine the factors influencing storytelling and behavioral intention in the context of RBCS by focusing on experiential value and emotions, which were identified in the literature. This research design is different from that used in previous studies on experiential value [18–20], which mainly addressed human services in the hospitality industry. The results will reveal key factors that predict behavioral intentions in consumers using RBCS. Based on the results, this study proposes practical solutions that are useful for developing a sustainable management in the RBCS context. This study also provides the fundamental information required to promote humanoid robot services in the hospitality industry in the post-COVID-19 era.

## 2. Literature Review and Hypotheses

### 2.1. Experiential Value

The concept of consumption experience is described in a broad sense by Lewis and Chambers [33] as the overall outcome of the products, services, and consumption environments encountered by consumers [34]. Experiential value refers to the perceived value that a consumer derives from direct use or indirect observation of a product or service [35]. Experiential value refers to customers' perceived value associated with their consumption experiences [18,19]. It is also defined as customers' perceived and relativistic preference for product attributes or service performances provided to consumers [16]. Thus, experiential values can be interactive, relative, preferred, personalized, and dynamically changed as experience accumulates [36,37].

Holbrook [36] proposed an experiential value typology suggesting different forms of values that are divided into eight quadrants reflecting three axis dimensions (i.e., intrinsic versus extrinsic, active versus reactive, and self-oriented versus other-oriented). Based on this, Mathwick et al. [16] developed an experiential value scale and conceptualized experiential value with dimensions of consumer return on investment (CROI), service excellence, aesthetics, and playfulness. First, CROI refers to the active investment of financial, temporal, behavioral, and psychological resources that can potentially yield a return [16]. It is derived from consumer spending and reflects a perceived rational quality of consumption experiences completed by evaluating economic utility of a good/service and by using time efficiently [16]. Favorable evaluations of CROI enhance the perceived deal value of an exchange and thereby help consumers feel good about their purchasing decisions [38].

Jin et al. [19] found that the emotional response to the service provider was positively affected by the perception of a reasonable price for the service experience. Alan et al. [34] confirmed CROI as a construct factor of experience-related cognitions and identified its positive association with positive emotion. Lacap [23] found that CROI has a direct positive association with positive emotion. Kim and Stepenkova [18] studied the relationship between experiential value and emotions in Korean family restaurants, finding that CROI had a positive relationship with positive emotion. Therefore, the following hypothesis is proposed.

**Hypothesis 1 (H1).** CROI has a significant positive effect on positive emotion.

Second, service excellence refers to an extrinsic value reflecting consumers' generalized perception of a service provider that demonstrates expertise and reliability or product performance [16]. This type of experiential value has potential to generate favorable feedback or emotions [18,34]. Gracia et al. [39] ascertained that consumers experience positive emotion such as happiness, joy, and excitement when they believe that a high-

quality service is being provided. Specifically, interaction with various factors of food and service in the food service industry can cause favorable or unfavorable feelings [40].

Jang and Namkung [41] found that the quality of excellent service had a significant positive effect on positive emotion in customers who visited full-service restaurants. Alan et al. [34] identified service excellence as a construct factor of experience-related cognitions for customers who visited coffee shops and confirmed its relationship with positive emotion. Therefore, the following hypothesis is proposed.

**Hypothesis 2 (H2).** Service excellence has a significant positive effect on positive emotion.

Third, atmosphere is closely related to aesthetics and is defined as a response to conformity, performance, and harmony of a physical thing, which is triggered by visual elements in the environment [16]. Separately, it is also described as visual attractiveness, such as the design of the physical environment [42]. The aesthetic value of the physical environment, such as atmosphere, can lead consumers to experience positive emotion [43]. A perceived good atmosphere is likely to generate feelings of pleasantness and service satisfaction among other positive emotion [44]. Jang and Namkung [41] confirmed that the atmosphere of full-service restaurants shows a positive association with consumers' positive emotion. Therefore, the following hypothesis is proposed.

**Hypothesis 3 (H3).** Atmosphere has a significant positive effect on positive emotion.

Fourth, escapism reflects sensory stimulation, namely a relief from boredom or stress based on getting away from the routine of daily life [45]. It can serve as an important factor that ultimately attracts new customers by allowing them to feel playfulness or other emotional values through escapism [17,18]. Hence, the value a customer places on escapism directly influences the symbolic meaning of a product/service, as well as the emotional arousal it instigates [46].

A study by Kim and Stepchenkova [18] confirmed the positive association of escapism with positive emotion. Alan et al. [34] identified escapism as a component of consumers' experience-related cognitions and confirmed its positive association with positive emotion. Therefore, the following hypothesis is proposed.

**Hypothesis 4 (H4).** Escapism has a significant positive effect on positive emotion.

*2.2. Emotions*

Batra and Stayman [47] defined emotions as a subjective state of mind that influences an individual's choice of affective messages. Consumer emotions are one of the critical areas in consumer behavior research [48]. Consumer emotions are an emotional reaction or feelings elicited through the experience of partaking of a product or service [22]. Consumer emotions are defined as an emotional response to store environmental stimuli [49] and consists of 10 different emotions such as interest, joy, surprise, sadness, anger, disgust, contempt, fear, shame, and guilt [50]. Emotions caused by these external stimuli are accompanied by behavioral responses [51]. Consequently, emotions are not merely reactions to appraisals but also include tendencies to actions [52,53].

The role of emotions has been identified as one of the factors that influence consumer attitudes and behavior in many studies [54]. What truly moves human beings is emotions and, based on these emotions, well-crafted storytelling can be developed [30]. According to Manthiou et al. [30], positive emotion about cruise travel positively affected storytelling. In addition, Ladhari [55] found that positive emotion such as pleasure had a significant positive relationship with positive WOM. Therefore, the following hypothesis is proposed.

**Hypothesis 5 (H5).** Positive emotion has a significant positive effect on storytelling.

Given that a story represents emotions, positive emotion positively influence WOM, intention to repurchase, recommendation, and consumer satisfaction [56]. Similarly, positive emotion resulting from consumer experience of a restaurant is positively related to loyalty [18]. The positive emotional response to AI services in the hospitality industry has also been found to influence acceptance of AI devices [14]. Jang and Namkung [41] confirmed the positive and significant relationship between positive emotion and behavioral intention. In addition, the correlation analysis by Kabaday and Alan [56] of positive emotion and revisit intention showed a significant correlation. Therefore, the following hypothesis is proposed.

**Hypothesis 6 (H6).** Positive emotion has a significant positive effect on behavioral intention.

### 2.3. Storytelling

In the hospitality industry, consumer-led storytelling behavior involves sharing information in the form of stories through WOM communication [28]. As consumers tend to believe the accounts of their personal contacts (such as family, friends, and neighbors) more than advertising, storytelling is regarded as an important source of information [57]. Storytelling is more compelling to consumers than statistics or facts because it reframes messages in an easy-to-grasp format that anyone can relate to [58,59]. In storytelling, ways of expressing the stories can be inspirational, and combining different stories together represents a history or legend [60]; in this way, storytelling becomes a persuasive tool of expression for listeners [61]. Storytelling is defined as sharing knowledge or experiences through a story to deliver a complicated idea, concept, or casual relation [62]. When delivered as stories and accepted by listeners, consumer knowledge and experience contribute to forming favorable attitudes and purchase intentions [63]. Hence, WOM effects that result from storytelling can evoke positive emotion from customers, allowing them to develop an intention to revisit [56]. Storytelling is considered one of the effective ways to influence behavioral intention [28].

In terms of the hospitality industry, Manthiou et al. [30] reported a positive causal relationship between storytelling and repurchase intention in their study on luxury cruise services. Ahn et al. [27] also found storytelling to have a positive effect on behavioral intentions (in terms of repurchase intention, revisit intention, recommendation, and positive WOM) among people visiting international exhibitions. Therefore, the following hypothesis is proposed.

**Hypothesis 7 (H7).** Storytelling has a significant positive effect on behavioral intention.

### 2.4. Behavioral Intention

Behavioral intention is the individual's belief and will to represent the consumer as a specific future behavior after forming an attitude toward an object [64]. It also refers to the possibility of the next step of action that consumers will take in the future, such as recommending or re-purchasing [65]. Behavioral intention is often used as a decision of behavior (act), meaning an individual's response to an object based on individual beliefs and emotions [66]. In general, many studies have used a combination of behavioral intention and loyalty, because there are similar conceptual aspects to behavioral intention and loyalty [67]. However, while loyalty includes only positive intention, behavioral intention have not only positive but negative aspects [68]. Zeithaml and Bitner [69] suggested behavioral intention as positive WOM, recommendation to others, promotion of affection, re-use with others, and willingness to pay premium prices. Moreover, Choi et al. [70] suggested that behavioral intention can be divided into repurchase intention and revisit intention.

All the hypotheses were included in the theoretical model, which is depicted in Figure 1.

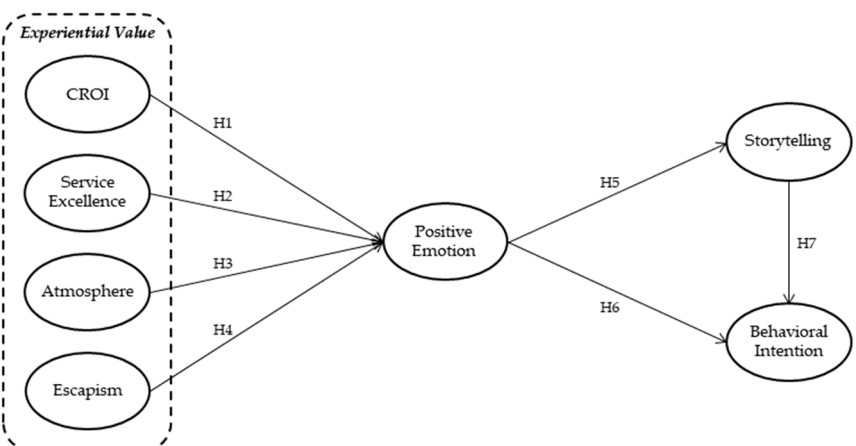

**Figure 1.** Research model.

## 3. Methodology

### 3.1. Research Instrument

In this study, questionnaire items were primarily structured based on a literature review and modified to accommodate for the experiences of RBCS. The modified questionnaire was reviewed by an expert group comprised of three professors in restaurant management and three RBCS owners to collect their opinion on the structure and content of the questionnaire. After the expert review, a pilot test was carried out to evaluate whether the items were clear and understandable to respondents. We also carried out an exploratory factor analysis (EFA) to finalize the questionnaire by removing questions related to immeasurable results and further clarifying the questions.

The questionnaire dataset has eight dimensions: CROI, service excellence, atmosphere, escapism, positive emotion, storytelling, and behavioral intention. Four items for CROI were adapted from Kim and Stepchenkova [18], Taylor et al. [20], and Tsai and Wang [37]. Three items for service excellence were derived from Rezaei and Valaei [71], Taylor et al. [20], and Tsai and Wang [37]. The pre-existing items were also used for atmosphere (*n* = 5) and escapism (*n* = 4), based on the work of Kim and Stepchenkova [18] and Tsai and Wang [37], respectively. Three items for positive emotion came from Kim and Stepchenkova [18]; four items for storytelling were obtained from Ahn et al. [27]; and three items for behavioral intention were obtained from Keng et al. [42], Taylor et al. [20]. All items were answered using a 5-point Likert scale ranging from "strongly disagree" to "strongly agree."

### 3.2. Pilot Test

EFA was performed on the 16 items in the pilot test to explore the dimensionality of the experiential value measurement. The pilot test was conducted to evaluate the 16-item questionnaire via online SNS using a convenience sample. To be selected for the study, potential participants were initially asked two screening questions regarding the definition of RBCS and whether they had visited RBCS within the last three months.

Data collected from 110 SNS users met the inclusion criteria and were used in the analysis. This sample size is considered sufficient on the basis of the minimum sample size of 100 and a minimum subject-to-item ratio of 5:1, as recommended by Gorsuch [72]. Of the 110 respondents, 23 (20.9%) chose "Café AI" as the RBCS that they mainly visited, accounting for the largest proportion, followed by Lounge'X (20%), Dalkomm b;eat (15.5%), and Coffee Method (11.8%). The remaining RBCSs accounted for less than 10%, respectively.

Subsequently, a principal component analysis was conducted with varimax rotation to determine whether the 16 items that are related to experiential values had loading on expected factors. Any items with factor loadings less than 0.4 and overall construct reliability below the acceptable value of 0.7 were removed [73]. After deleting some items, we reanalyzed the dataset. This process removed all three items of service excellence that did not fit well, and produced a three-factors solution containing 12 items, which

together accounted for 65.75% of the total variable [74]. The first factor included four components related to CROI ($\alpha = 0.721$), the second factor included five components related to atmosphere ($\alpha = 0.786$), and the third factor included four components related to escapism ($\alpha = 0.821$).

### 3.3. Sampling and Data Collection

In this study, a sample was extracted from the population of people aged 20 or older living in South Korea who had experience of visiting a RBCS in the last three months. Through the pilot test, the identified RBCS most frequently used by coffee consumers were identified for the survey—Dal.Komm b; eat, Café AI, Lounge'X, and Coffee Method. Therefore, the users of these four coffee shops are considered to represent the population of people using RBCS.

Online data collection was conducted by ENTRUST from 6 January to 20, 2021. EN-TRUST is a reliable global online research company based in South Korea and Hong Kong with a panel of over 560,000 people. Two screening questions were administrated before a respondent was invited for an interview. Have you visited one of "Café AI, Lounge'X, Dal.Komm b;eat, and Coffee Method" within the past three months? Do you know if "Café AI, Lounge'X, Dal.Komm b;eat, and Coffee Method" is a RBCS? The survey was terminated if any of the two answers was no. Of the 2830 participants who responded to the survey invitation, 300 participants answered "yes" to two questions, and they were briefed on the purpose and scope of the study. Their consent was obtained before initiating study procedures. Participants were asked if they had experience visiting one of the four RBSCS selected and if they had ordered something prepared by a robot barista in the last three months. Participants who met these inclusion criteria were briefed on the objectives of this study, and informed consent was obtained from them prior to the study. The questions were presented in the same order to all respondents, and a sample of 300 participants was used for empirical analysis. Table 1 presents a profile of the sample.

**Table 1.** Respondents' profiles.

| Demographic Characteristics | | Frequency | Percentage |
|---|---|---|---|
| Gender | Male | 140 | 46.7 |
| | Female | 160 | 53.3 |
| Age | 20–29 years | 71 | 23.7 |
| | 30–39 years | 117 | 39.0 |
| | 40–49 years | 63 | 21.0 |
| | 50–59 years | 31 | 10.3 |
| | Above 60 years | 18 | 6.0 |
| Marital status | Single | 134 | 44.7 |
| | Married | 166 | 55.3 |
| Educational level | High school | 19 | 6.3 |
| | 2-year university | 30 | 10.0 |
| | 4-year university | 223 | 74.3 |
| | Graduate school | 28 | 9.3 |
| Monthly income | Below USD 2000 | 26 | 8.7 |
| | USD 2000–2999 | 46 | 15.3 |
| | USD 3000–3999 | 57 | 19.0 |
| | USD 4000–4999 | 48 | 16.0 |
| | USD 5000–5999 | 40 | 13.3 |
| | USD 6000–6999 | 30 | 10.0 |
| | Above USD 7000 | 53 | 18.0 |

### 3.4. Analytical Methods

A statistical analysis was conducted using SPSS 22.0 and AMOS 22.0. The demographic characteristics of respondents were analyzed based on the data collected using SPSS 22.0. Data analysis was performed using Anderson and Gerbing [75]'s two-step approach to test our hypotheses: measurement model and structural model evaluation. Confirmatory factor

analysis (CFA) was first performed to test the adequacy of the measurement model and assess composite reliability, convergent validity, and discriminant validity. Subsequently, we performed the structural equation modelling to test hypothetical relations among the six proposes constructs.

## 4. Results

### 4.1. Measurement Model

The goodness-of-fit of the measurement model was assessed using CFA. Seven common model fit measures such as $\chi^2/df$ (<3), goodness-of-fit index (GFI > 0.9), root mean square error of approximation (RMSEA < 0.08), root mean square residual (RMR < 0.08), normed fit index (NFI > 0.9), incremental fit index (IFI > 0.9), and comparative fit index (CFI > 0.9) were used to estimate the measurement model fit [76]. Table 2 indicates the results of the CFA after removing one item of CROI and one item of atmosphere which reduced the goodness of fit of the model based on the squared multiple correlations (SMC > 0.4) value. The measurement model had a good fit with the data collected ($\chi^2$ = 284.043, df = 169, *p* = 0.000, CMIN/df = 1.681, RMR = 0.028, GFI = 0.916, NFI = 0.914, IFI = 0.963, CFI = 0.963, RMSEA = 0.048). The adequacy of the measurement model was assessed on the basis of reliability and convergent and discriminant validity. First, reliability was assessed based on composite reliability (CR) values. As shown Table 2, all the values exceed 0.7, demonstrating adequate CR [76]. The average variance extracted (AVE) values for all variables were higher than the proposed threshold of 0.5, indicating the convergence validity of the scale [76].

**Table 2.** Measurement model assessment.

| Variables & Item | SL | CR | AVE |
|---|---|---|---|
| CROI (Cronbach's α = 0.684) | | | |
| Using the RBCS has good economic value | 0.625 | 0.759 | 0.513 |
| Using the RBCS is a more efficient way to manage my time than using a general coffee shop | 0.686 | | |
| Using the RBCS is convenient | 0.637 | | |
| Atmosphere (AP) (Cronbach's α = 0.740) | | | |
| The robot barista's appearance is very impressive | 0.651 | 0.851 | 0.533 |
| The atmosphere in the RBCS is wonderful | 0.620 | | |
| The RBCS doesn't just sell coffee menus—it entertains me | 0.707 | | |
| I enjoyed watching the robot barista make a coffee menu | 0.620 | | |
| Escapism (EP) (Cronbach's α = 0.834) | | | |
| Using the RBCS releases me from reality and helps me truly enjoy myself | 0.757 | 0.851 | 0.590 |
| Using the RBCS makes me feel like I am in another world | 0.642 | | |
| I can relax my mood here | 0.762 | | |
| The experience at the RBCS was truly a joy | 0.773 | | |
| Positive emotion (PE) (Cronbach's α = 0.709) | | | |
| I was happy while using the RBCS | 0.693 | 0.760 | 0.613 |
| I was comfortable while using the RBCS | 0.647 | | |
| I was excited while using the RBCS | 0.559 | | |
| Storytelling (ST) (Cronbach's α = 0.808) | | | |
| I will post photos or videos of the RBCS on my SNS | 0.633 | 0.850 | 0.588 |
| I will tell my story about the RBCS to close friends and relatives | 0.660 | | |
| I will show photos of the RBCS to others | 0.783 | | |
| I will show videos of the robot barista's work to others | 0.812 | | |
| Behavioral intention (BI) (Cronbach's α = 0.820) | | | |
| I have a strong desire to visit the RBCS | 0.782 | 0.890 | 0.670 |
| I would more frequently visit the RBCS in the future | 0.831 | | |
| I would recommend the RBCS to friends | 0.783 | | |

Note: SL = standard loading, CR = composite reliability, AVE = average variance extracted.

To examine the discriminant validity of variables, for which convergent validity has been established, we compared the square root of the AVE of each latent variable against its corresponding correlation coefficient between latent variables. Table 3 shows that the square root of the AVE of each latent variable is greater than its corresponding correlation coefficient, implying adequate discriminant validity [77].

**Table 3.** Correlations of analysis between the variables.

| Variable | 1 | 2 | 3 | 4 | 5 | 6 |
|---|---|---|---|---|---|---|
| 1. CROI | 0.716 | | | | | |
| 2. AP | 0.341 | 0.744 | | | | |
| 3. EP | 0.421 | 0.454 | 0.766 | | | |
| 4. PE | 0.368 | 0.528 | 0.466 | 0.746 | | |
| 5. ST | 0.314 | 0.430 | 0.391 | 0.499 | 0.767 | |
| 6. BI | 0.378 | 0.393 | 0.421 | 0.528 | 0.473 | 0.837 |
| Mean | 3.835 | 3.964 | 3.598 | 3.915 | 3.377 | 3.753 |
| S.D. | 0.654 | 0.583 | 0.744 | 0.574 | 0.709 | 0.743 |

Note: Square root of AVE are on the diagonal. Squared correlations are below the diagonal.

### 4.2. Structural Model

A SEM was conducted using the AMOS 22.0 statistical package. To test the hypotheses established through the SEM path coefficients, the fit of the structural model describing the relationships among constructs was assessed. The model fit indices were $\chi^2$ = 278.562, df = 173, $p$ = 0.000, CMIN/df = 1.610, RMR = 0.027, GFI = 0.917, NFI = 0.916, IFI = 0.966, CFI = 0.966, and RMSEA = 0.045, thereby meeting the standard assessment criteria. The result of each hypothesis test describing the causal relationship between any pair of constructs is presented in Table 4. H1 was supported because CROI positively and significantly influenced positive emotion ($\beta$ = 0.293, $t$ = 2.031, $p$ = 0.042); H3 was supported because atmosphere positively and significantly influenced positive emotion ($\beta$ = 0.585, $t$ = 3.815, $p$ = 0.000); H4 was rejected because escapism did not significantly influence positive emotion ($\beta$ = 0.128, $t$ = 0.731, $p$ = 0.465); H5 was supported because positive emotion positively and significantly influenced storytelling ($\beta$ = 0.863, $t$ = 9.633, $p$ = 0.000); H6 was supported because positive emotion positively and significantly influenced behavioral intention ($\beta$ = 0.681, $t$ = 5.792, $p$ = 0.000); and H7 was supported because storytelling positively and significantly influenced behavioral intention ($\beta$ = 0.273, $t$ = 2.472, $p$ = 0.013).

**Table 4.** Results of the structural model analysis.

| Hypotheses | | $\beta$ | $t$-Value | $p$-Value | Decision |
|---|---|---|---|---|---|
| H1 | CROI → PE | 0.293 | 2.031 * | 0.042 | supported |
| H3 | AP → PE | 0.585 | 3.815 ** | 0.000 | supported |
| H4 | EP → PE | 0.128 | 0.731 | 0.465 | rejected |
| H5 | PE → ST | 0.863 | 9.633 ** | 0.000 | supported |
| H6 | PE → BI | 0.681 | 5.792 ** | 0.000 | supported |
| H7 | ST → BI | 0.273 | 2.472 * | 0.013 | supported |

Note 1: H2 from the original model was dropped as the factor service excellence was dropped in the principal component analyses. Note 2: * $p$ < 0.05, ** $p$ < 0.01.

## 5. Discussion

The data analysis showed that atmosphere had the greatest influence on customer emotions of RBCS among the factors of experience value. This result is consistent with previous studies [34,41]. Specifically, a combination of digitally advanced robot baristas' features, such as appearance and functions, coffee shop atmosphere, and entertainment elements boost positive emotion. CROI is also positively associated with positive emotion after experiencing an RBCS. This result is consistent with previous studies [18,19,23,34]. At the same time, convenience and prompt services provided by robot baristas are regarded as efficient, exerting a positive impact on customer emotions; a perceived economic value on the money spent to purchase goods and services in a RBCS also increases customers' positive emotion. However, escapism is found to have no significant effect on positive emotion. This means that the use of robot baristas is a happy experience, but not as happy as an escape from daily routine. These results are partially contrasts with previous studies [18,34] on experiential value. Kim and Stepchenkova [18] confirmed that there was no influence of atmosphere on positive emotion because customers took the attractive

dining environment of a westernized family restaurant for granted. However, the current study confirmed that customers who had little or no experience with the RBCS increased their positive emotion toward the RBCS by experiencing a better atmosphere than they expected. In addition, the relationship between escapism and positive emotion perceived by customers in RBCSs was different from those reported by Kim and Stepchenkova [18]. This means that the RBCS lacks elements that can provide a happy experience or sufficient relaxation to escape from everyday life. This is an important result that can confirm the emotional response of customers to human and robot service in the hospitality industry.

Positive emotion is found to increase customer-led storytelling about robot baristas. This result supports previous studies [30,55] on the relationship between positive emotion and storytelling. This also signifies that the feelings of pleasure and excitement evoked by the experience in a RBCS are important antecedents of storytelling. Positive emotion was found to increase behavioral intention toward RBCS, consistent with previous studies [41,56]. Also, storytelling was found to increase behavioral intention, consistent with previous studies [27,30]. These results confirm the role of positive emotion and storytelling in the formation of customer behavioral intentions in non-face-to-face service contexts.

### 5.1. Theoretical Implications

This study makes some important contributions to the literature. In many studies, experiential value has been cited as a customer's perceived value derived from their consumption experience. Existing experiential value studies have largely focused on human services [18–21]. Hence, little research has been done into consumer experience and the emotions associated with contactless services despite digital transformation and fast innovation. To cope with this situation, this study tests customers' emotional responses to robotic services by examining the causal relationship between experiential values and emotions. This design is different from previous studies on robot service in the hospitality industry [9,78,79]. Furthermore, this study has also identified the relationship between experiential value and emotions for robot services for the first time.

This study tested the relationship between factors and identified the most important antecedents of the emotions of customers who experienced RBCS. In addition, the results confirmed that the experiential value theory can be applied to the robot service context. However, unlike the human service context, service excellence could not measure the relationship with emotions in the robot service context. This is because customers perceive the service provided by the robot differently from the human service. As a result, we confirmed that only CROI and atmosphere among the sub-factors of experience value in the robot service context were related to customer's positive emotion. The results suggest that the experience value of the RBCS can increase storytelling and behavioral intention by appealing to customers' positive emotion. This confirms the importance of positive emotion within the model. This contributed to the hospitality literature as the results are different from the experiential value studies [18–21] that have been mainly applied to human services.

### 5.2. Practical Implications

From a practical point of view, experiential values and positive emotion are important factors influencing customer-led storytelling and behavioral intention in the RBCS context. Consumers are found to feel pleasure and interest in the RBCS atmosphere and be comfortable with the convenience, prompt services, and economic value, thereby building positive emotion. This suggests the need for developing "Soulware" robots, which are able to enhance intimacy with humans with the ability to understand human emotions and make different decisions depending on the situation, based on their intelligence and sensibility. "Soulware" robots that have "Human Touch" will soon be used to offer customized services according to a consumer's emotion, gender, age, taste, and specific situation [80]. In addition, robot baristas are required to provide consumers with a variety of products and services such as roastery, latte art, and hand drip in addition to serving menu items.

Robots' multiple functions will attract consumer attention and interest, boosting the positive emotion of consumers using the RBCS. If a seamless service is provided as a means of improving CROI, consumers will have better experiences, as the processing time for every order is shortened (i.e., convenience and promptness is increased). Robot baristas equipped with AI technology and deep learning algorithms, which are used for hyper personalization, will increase consumers' positive emotion while relieving their anxiety. This function of humanoid robot baristas will allow consumers to have higher expectations of the service in the future. At the same time, interesting robotic performances and games for entertainment and promotional events in an RBCS will help consumers relieve their daily stress and eventually increase potential positive emotion associated with RBCS.

Enjoyment and interesting feelings from the experience of RBCS can be shared with others through WOM. With the recent spread of contactless shopping, smart devices play a role in promoting non-contact culture and SNS communication. In particular, online storytelling has steeply increased in various forms, including posting comments and inserting links to create viral marketing effects. Hence, user friendly environments (e.g., photo zones and VR virtual spaces) in RBCSs that are designed to promote digital experiences and separate spaces for live shows, promotional videos, and promotional contents can encourage storytelling and attract more SNS users, further contributing to the intention to recommend and revisit.

Amid the COVID-19 pandemic, food tech and digital transformation have rapidly advanced, and consumers have begun to experience the convenience and efficiency of contactless services. In line with this trend, food technology experiences a transition from industrial to humanoid robots. In both online and offline markets, „Human Touch" (i.e., human sensibility) that can interact with consumers is one of the key factors influencing consumer purchase decisions. The application of „Human Touch" robotic services will become essential to enable innovative consumer experiences through mutually supportive human-robot interactions, and to satisfy customers in the food service industry.

## 6. Conclusions and Limitations

In response to the rapid growth in AI adoption as a key component of the 4th industrial revolution, this study determined the structural relationship between variables that are related to customer experiential values and emotions, storytelling, and behavioral intention in the RBCS context, while also verifying the adequacy of the model developed. Our research design and results contribute to the literature on robot services in the hospitality industry. Therefore, the theoretical framework we proposed and tested using this model can be applied to future studies that explore contactless services and customer experience in the post-COVID-19 food service industry. As the pandemic continues to accelerate digital transformation, increased social attention is necessary. In this regard, this study provides useful insight into the management of sustainable RBCS.

This study has the following limitations and suggested directions for future studies. First, as the survey on consumer experience of RBCSs was conducted online due to COVID-19, we could not explicitly provide measures of emotional responses experienced after service encounters. Second, since online surveys can induce selection bias [81], various methods need to be used to increase response rates in future studies. Third, there are some limitations in applying the research results to the entire hospitality industry, as this was an empirical study targeting RBCS.

**Author Contributions:** S.-R.Y. and H.-M.J. conceived and designed the experiments; H.-M.J. performed the experiments and analyzed the data; S.-R.Y., S.-H.K. and H.-M.J. wrote the paper. All authors have read and agreed to the published version of the manuscript.

**Funding:** This research received no external funding.

**Institutional Review Board Statement:** Not applicable.

**Informed Consent Statement:** Not applicable.

**Data Availability Statement:** The data presented in this study are available on request from the corresponding author.

**Conflicts of Interest:** The authors declare no conflict of interest.

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
