# Peer review of "How Does Experiential Value toward Robot Barista Service Affect Emotions, Storytelling, and Behavioral Intention in the Context of COVID-19?"

_sustainability, doi:10.3390/su14010450_

Round 1

Reviewer 1 Report

Dear author/authors,

The article addresses an interesting topic and is well structured allowing the capitalization of the results of the research undertaken, but some improvements are needed, as follows:

  • it would be appropriate to present the demographic characteristics of the sample in a tabular form;
  • also, the implications of these demographics should be considered in terms of the extent to which respondents' gender, age, or education are relevant in the research context;
  • the possibilities of using robots in other domains should also be discussed, as well as the feasibility of adopting this variant post-pandemic.

Author Response

Response to Reviewer 1 Comments

* The revised contents of the manuscript are presented in red.

Point 1: It would be appropriate to present the demographic characteristics of the sample in a tabular form; also, the implications of these demographics should be considered in terms of the extent to which respondents' gender, age, or education are relevant in the research context;

Response 1: Thank you very much for your comment. As you suggested, we present the demographic characteristics of the sample in tabular form.

Point 2: the possibilities of using robots in other domains should also be discussed, as well as the feasibility of adopting this variant post-pandemic.

Response 2: Thank you very much for your comment. We present the relevant content as a limitation at the end of the 'Conclusion and limitations' section.

We hope that our revised paper adequately meets your requirements and is worthy of being published in your esteemed journal.

Reviewer 2 Report

This is a good paper and has potential to attract good readership. The introduction reads well, authors however need to provide some further support for their hypotheses. Some relevant references on emotions and Covid-19 are: 

Ramkissoon, H. (2020). Perceived social impacts of tourism and quality-of-life: A new conceptual model. Journal of Sustainable Tourism, 1-17.

Ramkissoon, H. (2021). Place Affect Interventions During and After the COVID-19 Pandemic. Frontiers in Psychology, 3864.

The findings need to be further linked to literature. Overall this is a good paper, I recommend revisions. Thank you.

Author Response

Response to Reviewer 2 Comments

The revised contents of the manuscript are presented in red.

Point 1: This is a good paper and has potential to attract good readership. The introduction reads well, authors however need to provide some further support for their hypotheses.

Response 1: Thank you for your comment. We added some relevant references on emotion to support our hypotheses.

We hope that our revised paper adequately meets your requirements and is worthy of being published in your esteemed journal.

Reviewer 3 Report

It is a pleasure to read this paper which studied how experiential value (CROI, Service Excellence, Atmosphere, Escapism) impacts positive emotion and how positive emotion influences customers’ storytelling and behavioral intention.

Generally, this paper did solid work and presents a refreshing study of robot barista coffee shops. For the better improvement of this paper, I stress the following concerns.

COVID-19 Context:

  • In the introduction, the authors state that the outbreak of COVID-19 enables the growing use of robots in coffee shops and mention this study “establish a long-term relationship with customers in the RBCS context”. These statements should be very careful because the COVID-19 pandemic is a sudden disaster and it is still unknown how it will change business modes eventually.
  • In measurement, it is better to ask more COVID-19 related questions. Such as the emotions under COVID-19, “Using the RBCS has good economic value in the time of COVID”, etc.

Conceptual:

  • The authors mention there are both positive and negative emotions and behavioral intentions. Why focus on only positive sides toward RBCS? The perceived usefulness and perceived ease of use regarding robots will induce both positive and negative emotions. Especially, during COVID-19, people are negative enough even encountering novel technologies won’t trigger their happiness. This may also be the explanation for your insignificance of escapism. Please clarify.
  • Similarly, is there any negative storytelling?
  • How are results in robot barista coffee shops applicable to all businesses in the hospitality industry?

Details:

  • In line 202, the authors argue that “Emotions are therefore more persuasive than cognitive messages in terms of change of consumer behavior”. There is no priority on the importance of emotion and cognition. I also didn’t see a similar expression in your cited reference.
  • In line 445, I am not sure this paper is the first to define the relationship between experiential value and emotion firstly, maybe illustrate the context?
  • The writing should focus more on the topic without redundant concepts and sentences should be more logically connected.

GOOD LUCK!

Author Response

Response to Reviewer 3 Comments

* The revised contents of the manuscript are presented in red.

Point 1: In the introduction, the authors state that the outbreak of COVID-19 enables the growing use of robots in coffee shops and mention this study “establish a long-term relationship with customers in the RBCS context”. These statements should be very careful because the COVID-19 pandemic is a sudden disaster and it is still unknown how it will change business modes eventually.

Response 1: Thank you very much for your comment. We have removed the sentence accordingly.

Point 2: In measurement, it is better to ask more COVID-19 related questions. Such as the emotions under COVID-19, “Using the RBCS has good economic value in the time of COVID”, etc.

Response 2: Thank you for your comment. In future research, we will definitely use the measurement you suggested.

Point 3: The authors mention there are both positive and negative emotions and behavioral intentions. Why focus on only positive sides toward RBCS? The perceived usefulness and perceived ease of use regarding robots will induce both positive and negative emotions. Especially, during COVID-19, people are negative enough even encountering novel technologies won’t trigger their happiness. This may also be the explanation for your insignificance of escapism. Please clarify.

Response 3: Thank you for your comment. If the relationship between the sub-factors of experience values and positive emotions is identified, then the direction of influence may be positive, negative, or there may be no influence. Therefore, we applied only one positive emotion factor to the model because positive and negative aspects can be verified with only positive emotions.

Point 4: Similarly, is there any negative storytelling?

Response 4: Thank you for your comment. We could not find any negative storytelling factors in previous studies. In addition, we determined that the positive and negative aspects could be verified with the storytelling factors applied in this study, as in the case of positive emotions.

Point 5: How are results in robot barista coffee shops applicable to all businesses in the hospitality industry?

Response 5: We present the relevant content as a limitation at the end of the 'Conclusion and limitations' section.

Point 6: In line 202, the authors argue that “Emotions are therefore more persuasive than cognitive messages in terms of change of consumer behavior”. There is no priority on the importance of emotion and cognition. I also didn’t see a similar expression in your cited reference.

Response 6: Thank you for your comment. We removed the sentence you pointed out and added a new one. We will take a more caution in the future when citing previous studies.

Point 7: In line 445, I am not sure this paper is the first to define the relationship between experiential value and emotion firstly, maybe illustrate the context?

Response 7: We corrected the sentence to "Furthermore, this study has also identified the relationship between experiential value and emotions for robot services for the first time" by adding the missing word "for robot services" in the sentence you referred to.

We hope that our revised paper adequately meets your requirements and is worthy of being published in your esteemed journal.

Reviewer 4 Report

  1. I suggest adding some international references (mainly new books focused on Covid-19 published by Routledge or other international Publishing Houses) to the article that can enrich the content.
  2. Generally, the article is high-quality work, from the organizational and scientific site.

Author Response

Response to Reviewer 4 Comments

* The revised contents of the manuscript are presented in red.

Point 1: I suggest adding some international references (mainly new books focused on Covid-19 published by Routledge or other international Publishing Houses) to the article that can enrich the content.

Response 1: Thank you for your comment. We cited articles published by other international publishing houses as per your suggestion.

We hope that our revised paper adequately meets your requirements and is worthy of being published in your esteemed journal.
